# Co-exposure maximization in online social networks

**Sijing Tu**
Department of Computer Science
KTH Royal Institute of Technology
Stockholm, Sweden
sijing@kth.se

**Cigdem Aslay**
Department of Computer Science
Aarhus University
Aarhus, Denmark
cigdem@cs.au.dk

**Aristides Gionis**
Department of Computer Science
KTH Royal Institute of Technology
Stockholm, Sweden
argioni@kth.se

## Abstract

Social media has created new ways for citizens to stay informed on societal matters and participate in political discourse. However, with its algorithmically-curated and virally-propagating content, social media has contributed further to the polarization of opinions by reinforcing users' existing viewpoints. An emerging line of research seeks to understand how content-recommendation algorithms can be re-designed to mitigate societal polarization amplified by social-media interactions. In this paper, we study the problem of allocating seed users to opposing campaigns: by drawing on the equal-time rule of political campaigning on traditional media, our goal is to allocate seed users to campaigners with the aim to maximize the expected number of users who are co-exposed to both campaigns. We show that the problem of maximizing co-exposure is **NP**-hard and its objective function is neither submodular nor supermodular. However, by exploiting a connection to a submodular function that acts as a lower bound to the objective, we are able to devise a greedy algorithm with provable approximation guarantee. We further provide a scalable instantiation of our approximation algorithm by introducing a novel extension to the notion of random reverse-reachable sets for efficiently estimating the expected co-exposure. We experimentally demonstrate the quality of our proposal on real-world social networks.

## 1 Introduction

Social media have created new ways for citizens to stay informed and participate in societal discourse. However, despite enabling users to access a variery of information, social media has been linked to increased societal polarization [22], by amplifying the phenomenon of echo chambers [4, 27], where users are only exposed to information from like-minded individuals, and of filter bubbles [35, 37], where algorithms only present personalized content that agrees with the user's viewpoint. To address these concerns, an emerging line of research seeks to understand how content-recommendation algorithms can be re-designed to mitigate societal polarization amplified by social-media interactions. Recent work includes developing methods for balancing [25] and diversifying [3] information exposure, while considering the tendency of the recommended content to spread through the online social network under a stochastic information propagation model.

In this paper, we take a step in this direction and consider the problem of breaking filter bubbles through the information-propagation lens. Following related work that has considered the problem

of viral marketing for multiple items in online social networks [1, 2, 12, 20, 21, 29], we consider a setting with a centralized authority (*host*) that is responsible for allocating seed nodes to campaigns. We assume that two campaigns, supporting opposing sides of a controversial social issue, approach the host to benefit from its viral-marketing service. By drawing on the equal-time rule of political campaigning in the pre-digital era, our aim is to devise a seed-set allocation framework so that the expected number of users who are exposed to both campaigns, through the propagation of information in the social network, is maximized.

In a traditional viral-marketing setting, both the campaigners and host would be interested in reaching out to the maximum number of people without any consideration to co-exposure [1, 2, 12, 20, 21, 29]. However, the new era of fake news and polarization has brought to the fore the fact that, algorithmic principles that work well for commercial advertising have unintended consequences when applied to political advertising, due to their commercial focus that prioritize revenue, resulting in biased and imbalanced campaigning. Such adverse effects have led to social-media platforms being held accountable for having political bias in their services. In some cases, popular social media have stopped their political advertising service at the expense of losing revenue. Thus, we assume the host has an incentive to respect balance and objectivity considerations, due to enforced legislations or social conscience.

As a step of addressing the aforementioned challenges, we formally introduce *co-exposure maximization* (CoEM) as the problem of assigning seed sets to each campaign such that the expected number of users co-exposed to campaigns under a stochastic information propagation model is maximized. We show that the CoEM problem is **NP**-hard and **NP**-hard to approximate within a factor better than $1 - 1/e$. Although the co-exposure function is neither submodular nor supermodular, we propose a greedy algorithm that exploits a connection to a submodular function that acts as the lower bound of the objective and obtain bounded approximation guarantees. Due to the #**P**-hardness of expected spread computation, we introduce a novel extension to the notion of random reverse-reachable sets [9] for efficiently estimating the expected co-exposure. Finally, we experimentally evaluate our algorithm on several real-world datasets and demonstrate its superiority over several baselines.

Omitted proofs and implementation are provided as supplementary material.

## 2  Related work

Our work relates to the emerging line of research on breaking filter bubbles in social media through information-propagation lens. There have been a number of studies on the effects of "echo chambers" [4, 26] and "filter bubbles" [4, 17, 22, 37]. In particular, it has been observed that news stories containing opinion-challenging information spread less than other news [26] and filtering of content by a social-network owner to increase user engagement can significantly increase societal polarization [17]. Recent approaches to breaking filter bubbles focus on making recommendations to individuals of opposing viewpoints [23, 24, 31], targeting users so as to reduce the polarization of opinions and bridge opposing views by considering opinion-formation models [16, 33, 34], or addressing these issues under information-propagation models [3, 25] as we do in our work.

Aslay et al. [3] study the related problem of diversifying exposure to information that is propagating in a social network. Their problem formulation assumes that the leanings of users and news articles are quantified in the interval $[-1, 1]$ and are known. The goal is to find an assignment of articles to seed users to maximize the total diversity over all users in the network. The diversity of a user is defined to be a function that takes as input the leanings of the set of news articles that the user is exposed as well as the learning of the user. Therefore, users who are exposed to only one article, which has a different leaning from their own, still contribute to the value of the diversity objective. Translating this formulation to our setting, by considering two articles with leanings $-1$ and $1$, implies that their objective function can potentially achieve a relatively high value while the co-exposure being equal to $0$. Thus, their work does not guarantee that co-exposure is maximized. Moreover, Aslay et al. [3] also propose an extension to random reverse-reachable sets [9] for scalable estimation of their objective function. For this task, they sample random sets defined over user-article pairs while our sampling domain is user-user pairs as we explain in Section 5. Due to the difference in the objective functions, hence, the estimation task, the sample-complexity results and the sample of random sets obtained for one problem cannot be used to solve the other.

The work most related to ours is the one by Garimella et al. [25], in which they consider a similar information-propagation setting with two opposing campaigns. Different from our work, they assume a set of initial seed sets to be given for each campaign and aim to recruit additional seed sets to maximize the expected number of users exposed either to both or none of the campaigns. The seed sets are not required to be disjoint, which makes their setting less realistic, as the seeds of a campaign are more likely to take one-sided stance. Furthermore, among the three algorithms they propose, only one provides an approximation guarantee for the setting where campaigns have different propagation probabilities, while the other two algorithms rely on limiting assumptions, such as, campaigns having same propagation probabilities and choosing common seeds for both campaigns. Another difference with the current work is that the co-exposure function we consider here is arguably a more natural choice for reducing polarization, as it accounts solely for nodes that are informed by both sides of a controversial issue. Finally, we note that Garimella et al. [25] use computationally prohibitive Monte Carlo simulations while we propose an efficient co-exposure estimation framework.

## 3    Problem definition

**Ingredients.** The input to our problem consists of: ($i$) a directed social graph $G = (V, E)$ with $|V| = n$ nodes and $|E| = m$ edges, where a directed edge $(u, v)$ indicates that node $v$ follows node $u$, thus, $v$ can see and propagate posts by $u$; ($ii$) two campaigners, campaigning for opposing sides of a controversial issue, with their campaigns hereafter referred as campaign $r$ (red) and campaign $b$ (blue), and their seed set budgets denoted by $k_r \in \mathbb{Z}_+$ and $k_b \in \mathbb{Z}_+$, respectively; and ($iii$) campaign-specific propagation probabilities $p_{uv}^r$ and $p_{uv}^b$, for all $(u, v) \in E$, representing the probability that a post from node $u$ will propagate to node $v$ in the respective campaigns.

Given the budgets $k_r$ and $k_b$ of the campaigners, the host is in charge of selecting disjoint seed sets $S_r$ and $S_b$ for advertiser $r$ and $b$, respectively, while respecting their seed set budgets, i.e., $|S_r| \leq k_r$ and $|S_b| \leq k_b$. The goal, on a high level, is to select seeds for the two campaigns, within the allocated budgets, so as to maximize the number of nodes in the network who are exposed to both campaigns.

**Propagation model.** We assume that the propagation of each campaign follows the independent-cascade (IC) model [28], each with campaign-specific propagation probabilities $p_{uv}^r$ and $p_{uv}^b$, for all directed edges $(u, v) \in E$. We assume that the propagation of a campaign through the edge $(u, v)$ is independent of $v$'s activation status on the other campaign. This way, we are able to take into account the tendency of nodes to adopt information they agree or disagree with, and model a realistic setting where users can adopt more than one campaign. This is in contrast to the competitive propagation models that assume that a user can adopt only one campaign [7, 10, 15, 41, 32]. Thus, once a node $u$ becomes active at time $t$ on campaign $r$ (respectively, $b$), it has one shot to activate each inactive out-neighbor $v$ at time $t + 1$, with probability $p_{uv}^r$ (respectively, $p_{uv}^b$), independently of the history thus far. We say that a node $u$ is *exposed* to a specific campaign if $u$ is activated on that campaign, either by an in-neighbor that is active on the same campaign, or by directly being a seed node of the campaign.

**Possible-world semantics.** Given any two sets of seeds $S_r$ and $S_b$ for the two campaigns, a single *possible world* represents an outcome of the stochastic propagation processes starting from the nodes in $S_r$ and $S_b$. To formalize the *possible-world semantics* of our problem, we adopt the edge-colored multigraph representation introduced by Aslay et al. [3]. Accordingly, we define a *directed edge-colored multigraph* $\tilde{G} = (V, \tilde{E}, \tilde{p})$ from $G = (V, E)$, by creating a parallel edge $(u, v)_i$ associated with color $c_i$ and probability $p_{uv}^i$, for each campaign $i \in \{r, b\}$. This way, $\tilde{G}$ can be regarded as a probability distribution over all the possible subgraphs of $(V, \tilde{E})$. The probability of a possible world $w \sqsubseteq \tilde{G}$, obtained by sampling each $(u, v)_i$ independently with probability $p_{uv}^i$, is thus given by

$$\Pr[w] = \prod_{i \in \{r,b\}} \prod_{(u,v)_i \in w} p_{uv}^i \prod_{(u,v)_i \in \tilde{E} \setminus w} (1 - p_{uv}^i).$$

Let $I_w(S_r)$ (respectively, $I_w(S_b)$) denote the set of nodes that are reachable from the nodes in $S_r$ (respectively, $S_b$) in $w$, by using the edges associated with color $c_r$ (respectively, $c_b$). Let also $C_w(S_r, S_b) = |I_w(S_r) \cap I_w(S_b)|$ denoted to set of nodes co-exposed to both campaigns in $w$. Then, the expected number of nodes co-exposed to both campaigns is given by

$$\mathbb{E}[C(S_r, S_b)] = \sum_{w \sqsubseteq \tilde{G}} \Pr[w] \, |I_w(S_r) \cap I_w(S_b)|.$$

We are now ready to formally define the problem we study in this paper.

**Problem 3.1** (CO-EXPOSURE MAXIMIZATION (COEM)). *Given a directed social graph $G = (V, E)$, two opposing campaigns $r$ and $b$, campaign-specific propagation probabilities $p_{uv}^r$ and $p_{uv}^b$, for all $(u, v) \in E$, and two positive integers $k_r$ and $k_b$, find two disjoint seed sets $S_r$ and $S_b$, such that $|S_r| \leq k_r$ and $|S_b| \leq k_b$, and the expected number of nodes co-exposed to the $r$ and $b$ campaigns is maximized. That is,*

$$\underset{S_r, S_b \subseteq V}{\text{maximize}} \quad \mathbb{E}\left[C(S_r, S_b)\right]$$

$$\text{subject to} \quad S_r \cap S_b = \emptyset,$$
$$|S_r| \leq k_r,$$
$$|S_b| \leq k_b.$$

## 4  Theoretical analysis

In this section, we establish the complexity of the COEM problem, and we theoretically analyze its objective function. We start by showing that COEM is **NP**-hard to approximate within a factor better than $1 - 1/e$.

**Theorem 4.1.** *It is **NP**-hard to approximate the COEM problem within a factor better than $1 - \frac{1}{e}$.*

Next we proceed to analyze the properties of the objective of the COEM problem.

Our first observation is that when the seed set of one of the campaigns is fixed, e.g., $S_r$, the objective function $\mathbb{E}[C(S_r, S_b)]$ is submodular in $S_b$, and vice versa. This observation follows from the submodularity of $\mathbb{E}[I]$ under the independent-cascade model [28]. However, the COEM problem requires to evaluate $\mathbb{E}[C(S_r, S_b)]$ over the pair of seed sets. Bi-submodularity [38, 36] extends the concept of submodularity to bi-set functions, i.e., set functions with two arguments. We first show that $\mathbb{E}[C]$, which is a bi-set function, is not bi-submodular.

**Lemma 4.2.** *The function $\mathbb{E}[C] : 2^V \times 2^V \to \mathbb{Z}_{\geq 0}$ is a non-decreasing bi-set function, which is not bi-submodular.*

In the rest of the section, we will first provide an equivalent univariate formulation of our objective, by defining a function $f : 2^{\mathcal{E}} \to \mathbb{Z}_{\geq 0}$ on the ground set $\mathcal{E}$ of ordered pairs of nodes. We will then show that $\mathbb{E}[f]$ is neither submodular nor supermodular.

For non-decreasing functions that are not submodular, the greedy algorithm, although being useful in practice, might not provide theoretical performance guarantees. However, in cases that the function is close to being submodular, the performance of the greedy algorithm tends to improve. The deviation of a function from submodularity is typically captured by the *submodularity ratio* of the function [18, 19]. This observation has been used in the literature to show that the greedy algorithm enjoys a tight approximation guarantee of $\frac{1}{\gamma_c}(1 - e^{-\gamma_c \gamma_r})$ for maximizing a non-submodular function subject to a cardinality constraint, where $\gamma_r$ and $\gamma_c$ are the submodularity ratio and the generalized curvature of the non-submodular function, respectively [8].

We note that such efforts are dedicated to approximating a non-submodular set function under a *cardinality* constraint. As we will see next, the constraints of our problem form an independence system on the ground set $\mathcal{E}$ of ordered pairs of nodes. We also note that $\mathbb{E}[f]$ has a submodularity ratio of 0 for many problem instances, limiting the possibility of obtaining a bounded approximation when greedily maximizing the objective function. Hence, instead, we will exploit a connection of $\mathbb{E}[f]$ to a submodular function, that acts as its lower bound, to obtain bounded approximation guarantees.

We now start formalizing our approach built on a set system defined on ordered pairs of nodes, which we name as *set-of-pairs* system, and establish its equivalence to the pairs of seed sets.

Let $(S_r^*, S_b^*)$ denote the optimal pair of seed sets maximizing the expected co-exposure. Being a non-decreasing function, the maximum value of the objective function is attained when $k_r$ and $k_b$ seed nodes are selected for campaigns $b$ and $r$, respectively. This implies that $|S_r^*| = k_r$ and $|S_b^*| = k_b$.

Without loss of generality we assume that $k_r \leq k_b$. Let $O_1 = \{(S_r, S_b) \mid S_r \cap S_b = \emptyset, |S_r| = k_r, |S_b| = k_b, S_r, S_b \subseteq V\}$ denote the set of feasible seed set pairs of maximal size. Notice that $(S_r^*, S_b^*) \in O_1$. For any $(S_r, S_b) \in O_1$, it follows that $1 \leq \frac{|S_b|}{|S_r|} \leq \lceil \frac{k_b}{k_r} \rceil$. This implies that we can

construct a set of pairings of the nodes in $S_r$ and $S_b$ such that each element in $S_r$ corresponds to at least one and at most $\lceil \frac{k_b}{k_r} \rceil$ elements in $S_b$.

Let $\mathcal{E} = \{V \times V\} \setminus \{(v, v) \mid v \in V\}$ denote a ground set of ordered pairs, where $(u, v) \in \mathcal{E}$ represents the pairing of a node $u$, selected as a seed node for campaign $r$ with a node $v$ selected as a seed node for campaign $b$. We now formally define *set-of-pairs* system defined on $\mathcal{E}$, and establish its relation to $O_1$.

**Definition 4.1** (Set-of-pairs system). *Let $(\mathcal{E}, \mathcal{I})$ be a set system where $\mathcal{E} = \{V \times V\} \setminus \{(v, v) \mid v \in V\}$ is the ground set and $\mathcal{I}$ is a collection of subsets of $\mathcal{E}$. For any $X \in \mathcal{I}$, let $X_r = \bigcup\{r \mid (r, b) \in X\}$ and $X_b = \bigcup\{b \mid (r, b) \in X\}$. We say that $(\mathcal{E}, \mathcal{I})$ is a set-of-pairs system if for any set $X \in \mathcal{I}$ the following conditions hold: (i) $|X_r| \le k_r$; (ii) $|X_b| = |X| \le k_b$; (iii) $X_r \cap X_b = \emptyset$; (iv) for each $r_0 \in X_r$, $|\bigcup\{b \mid (r_0, b) \in X\}| \le \lceil \frac{k_b}{k_r} \rceil$.*

**Lemma 4.3.** *Let $O_2 = \{(X_r, X_b) \mid X \in \mathcal{I}\}$. Then $O_1 \subseteq O_2$.*

Let $f : 2^{\mathcal{E}} \to \mathbb{Z}_{\ge 0}$ be a function defined as $f(X) = |I(X_r) \cap I(X_b)|$ where $I(X_r)$ and $I(X_b)$ are the random variables representing the set of the nodes exposed to campaigns $r$ and $b$, respectively. Since $(S_r^*, S_b^*) \in O_1$, it follows from Lemma 4.3 that $(S_r^*, S_b^*) \in O_2$. Thus, the CoEM problem can be equivalently formulated as

$$\max_{X \in \mathcal{I}} \quad \mathbb{E}[f(X)]. \tag{1}$$

**Lemma 4.4.** *The function $f : 2^{\mathcal{E}} \to \mathbb{Z}$ defined above is a non-decreasing set function, which is neither submodular nor supermodular.*

Next, we give a formal definition of the submodularity ratio and formalize the problem instances in which $\gamma_r = 0$.

**Definition 4.2** (Submodularity ratio [8]). *The submodularity ratio of a non-negative set function $F$ is the largest scalar $\gamma_r$ such that*

$$\sum_{e \in X \setminus L} F(L \cup \{e\}) - F(L) \ge \gamma_r \left( F(L \cup X) - F(L) \right), \text{ for all } L, X \subseteq \mathcal{E}.$$

*The set function $F$ is submodular if and only if $\gamma_r = 1$.*

For each $u \in V$, let $\sigma(u)$ denote the set of nodes that node $u$ can reach in $G$. Assume that there exists at least two ordered pairs $(r_1, b_1)$ and $(r_2, b_2)$ in $\mathcal{E}$, where $\sigma(r_i) \cap \sigma(b_i) = \emptyset$, for $i = 1, 2$, and there exist $i, j \in \{1, 2\}$, with $i \ne j$, such that $\sigma(r_i) \cap \sigma(b_j) \ne \emptyset$. Let $L = \emptyset$ and $X = \{(r_1, b_1), (r_2, b_2)\}$. Notice that, in any realization of the stochastic propagation process starting from the seed sets $X_r$ and $X_b$, we have $f(\{(r_i, b_i)\}) = 0$, hence, $\mathbb{E}[f(\{(r_i, b_i)\})] = 0$. Moreover, given our assumption that $\sigma(r_i) \cap \sigma(b_j) \ne \emptyset$ in at least one of the cross pairings, we have $\mathbb{E}[f(X)] > 0$, yielding $0 \ge \gamma_r \mathbb{E}[f(X)]$, thus, $\gamma_r = 0$.

We have shown that $\mathbb{E}[f(X)]$ is not submodular, with a submodularity ratio of $0$ in some problem instances. We now introduce a function $g : 2^{\mathcal{E}} \to \mathbb{Z}_{\ge 0}$, defined as $g(X) = |\cup_{(r,b) \in X} (I(r) \cap I(b))|$, which we show is submodular and differs from $f(X)$ within a multiplicative factor in any realization of the stochastic propagation process.

**Lemma 4.5.** *The function $g : 2^{\mathcal{E}} \to \mathbb{Z}$ defined above is a submodular non-decreasing set function.*

Notice that submodularity of $g$ implies submodularity of $\mathbb{E}[g]$ since non-negative linear combination of submodular functions is also submodular. We now prove the relation between the optimal solutions maximizing $\mathbb{E}[f]$ and $\mathbb{E}[g]$ subject to $X \in \mathcal{I}$.

**Lemma 4.6.** *Let $X^0 = \arg\max_{X \in \mathcal{I}} \mathbb{E}[g(X)]$, $X^* = \arg\max_{X \in \mathcal{I}} \mathbb{E}[f(X)]$, and $k_r \le k_b$. Then $\mathbb{E}[f(X^*)] \le k_r \mathbb{E}[g(X^0)]$.*

Lemma 4.6 suggests that any algorithm that provides an approximation guarantee for the problem of maximizing $\mathbb{E}[g]$ over the set system $(\mathcal{E}, \mathcal{I})$, provides also a bounded approximation guarantee for the CoEM problem. We now study the properties of the set-of-pairs system $(\mathcal{E}, \mathcal{I})$. We first provide the preliminary definitions.

**Definition 4.3** (Independence system). *A set system $(\mathcal{E}, \mathcal{I})$ is an independence system if $\mathcal{I}$ is non-empty and satisfies the downward-closure property, i.e., $X \in \mathcal{I}$ and $Y \subseteq X$ imply $Y \in \mathcal{I}$.*

Algorithm 1: Pairs-Greedy

**Input** : $G = (V, E, p), (\mathcal{E}, \mathcal{I})$.
**Output** : $X^G$

1  $\mathcal{X}^G \leftarrow \emptyset$
2  **while** $\mathcal{E} \neq \emptyset$ **do**
3       $y = \arg\max_{x : X^G \cup \{x\} \in \mathcal{I}} \mathbb{E}[g(X^G \cup \{x\})] - \mathbb{E}[g(X^G)]$
4       $\mathcal{E} \leftarrow \mathcal{E} \setminus \{y\}$
5       $X^G = X^G \cup \{y\}$
6  **end**
7  **return** $X^G$

**Definition 4.4** ($p$-system [11]). *An independence system $(\mathcal{E}, \mathcal{I})$ is said to be a $p$-system, if*

$$\max_{Y \subseteq \mathcal{E}} \frac{\max_{J : J \text{ is a base of } Y} |J|}{\min_{J : J \text{ is a base of } Y} |J|} \leq p,$$

*where any subset $J$ of $Y$ is a base of $Y$ if $J \in \mathcal{I}$ and for all $e \in Y \setminus J$ it is $J \cup \{e\} \notin \mathcal{I}$.*

We now show that $(\mathcal{E}, \mathcal{I})$ is an independence system.

**Lemma 4.7.** *The set-of-pairs system $(\mathcal{E}, \mathcal{I})$ is an independence system.*

**Lemma 4.8.** *The independence system $(\mathcal{E}, \mathcal{I})$ is a $2\lceil \frac{k_b}{k_r} \rceil$-system.*

Combining the result of Lemma 4.8 that $(\mathcal{E}, \mathcal{I})$ is a $2\lceil \frac{k_b}{k_r} \rceil$-system with the monotonicity and sub-modularity of $\mathbb{E}[g]$, the greedy algorithm (Algorithm 1) provides an $\frac{1}{1 + 2\lceil \frac{k_b}{k_r} \rceil}$-approximation [11].

Let $X^G \subseteq \mathcal{E}$ denote the solution returned by Algorithm 1. Let also $X_r^G = \bigcup \{r \mid (r, b) \in X^G\}$ and $X_b^G = \bigcup \{b \mid (r, b) \in X^G\}$. We now show that, the pair $(X_r^G, X_b^G)$ of seed sets provides $1/((1 + 2\lceil \frac{k_b}{k_r} \rceil)k_r)$-approximation to the optimal solution of the CoEM problem.

**Theorem 4.9.** *Let $X^G$ be the solution returned by Algorithm 1 and let $X_f^G = \{(r, b) \mid r \in X_r^G, b \in X_b^G\}$ denote all possible ordered pairings between $X_r^G$ and $X_b^G$. Then*

$$\mathbb{E}[f(X_f^G)]) \geq \frac{1}{(1 + 2\lceil \frac{k_b}{k_r} \rceil)k_r} \mathbb{E}[f(X^*)].$$

## 5 Algorithms

Efficient implementation of the greedy algorithm is a challenge as the computation of the expected spread $\mathbb{E}[I(S)]$ for any given $S$ is a #**P**-hard problem under the independent-cascade model [14]. A common practice is to estimate the expected spread using Monte Carlo simulations [28]. However, accurate estimation requires a large number of Monte Carlo simulations, which is prohibitively expensive, and results in a time complexity of $O(k_b m n^2 r)$ when using $r$ rounds of Monte Carlo simulations in Algorithm 1.

Considerable effort has been devoted to developing scalable influence-maximization algorithms. Borgs et al. [9] made a breakthrough by introducing the notion of sampling *reverse-reachable* (RR) sets, and proposed a quasi-linear time randomized algorithm for the influence-maximization problem. Tang et al. improved it to a near-linear time randomized algorithm, called *two-phase influence maximization* (TIM) [39], and subsequently proposed IMM [40] with a tightened lower bound on the sample size required to estimate the expected spread with high probability.

Random RR-sets are critical for efficient estimation of the expected influence spread. However, they are designed for the standard influence-maximization problem. In this section, we introduce a non-trivial generalization of RR-sets, which we name *reverse-reachable pairs* sets (RRP-sets), and accordingly devise an estimator for accurate estimation of $\mathbb{E}[g(X)]$ for any $X \in \mathcal{I}$.

**Definition 5.1** (Random reverse-reachable pairs (RRP) set). *Let $w \sqsubseteq \tilde{G}$ be any possible world and let $v \in V$ be a node uniformly sampled at random. A random RRP-set $R \subseteq \mathcal{E}$ is defined as the set of*

*ordered pairs that can reach node $v$ via the colored edges in $w$, i.e.,*

$$R = \{(r, b) : v \in I_w(r) \cap I_w(b)\}.$$

A RRP-set can be sampled efficiently by first sampling a node $v \in V$ uniformly at random, then performing a randomized breadth-first search starting from $v$ in $\tilde{G}$ as follows. Let $N^{in}(v)$ denote the set of in-neighbors of node $v$ in $\tilde{G}$. Initially, for each campaign $i \in \{r, b\}$, create an empty breadth-first search queue $Q_i$, and for each $z \in N^{in}(v)$, insert $z$ into $Q_i$ with probability $p^i_{zv}$. The following loop is executed until $Q_i$ is empty: dequeue a node $u$ from $Q_i$ and examine its *incoming* edges of color $i$: for each edge $(w, u)_i$, insert $w$ into $Q_i$ with probability $p^i_{wu}$. Let $A_i$ denote the set of nodes dequeued from $Q_i$. Then, a RRP-set $R$ is constructed from the set of all possible ordered pairings between $A_r$ and $A_b$, i.e., $R = \{(r, b) \mid r \in A_r, b \in A_b\}$.

Let $\mathcal{R}$ denote a sample of random RRP-sets generated by the procedure described above. Given a sample $\mathcal{R}$, let $F_{\mathcal{R}}(X) = \sum_{R \in \mathcal{R}} \mathbb{1}[R \cap X \neq \emptyset]/|\mathcal{R}|$ denote the fraction of RRP-sets that have non-empty intersection with $X$. Next, we show that, for any given $X \subseteq \mathcal{E}$, we can estimate $\mathbb{E}[g(X)]$ by estimating $\mathbb{E}[F_{\mathcal{R}}(X)]$. We have the following.

**Lemma 5.1.** *For any $X \subseteq \mathcal{E}$, we have $\mathbb{E}[g(X)] = n\,\mathbb{E}[F_{\mathcal{R}}(X)]$, where the expectation is taken over the randomness in $v \sim V$ and $w \sim \tilde{G}$.*

We now present our algorithm TCEM (two-phase co-exposure maximization) that provides an approximate-greedy solution $\tilde{X}^G$ to the problem of maximizing $\mathbb{E}[g(X)]$ using a sample $\mathcal{R}$. The TCEM algorithm operates in two phases: $(i)$ sampling phase, which determines the size of the sample required for accuracy of estimations and generates the sample $\mathcal{R}$; $(ii)$ the greedy pair selection phase, where, at each iteration, a feasible pair maximizing $F_{\mathcal{R}}(\tilde{X}^G \cup \{x\}) - F_{\mathcal{R}}(\tilde{X}^G)$ is added to the solution $\tilde{X}^G$.

Next we show that, given a sample $\mathcal{R}$ of random RRP-sets from which we can obtain accurate estimations of $\mathbb{E}[F_{\mathcal{R}}(X)]$ for all $X \in \mathcal{I}$ with high probability, we can solve CoEM accurately with high probability. Let OPT $= \mathbb{E}[g(X^0)]$, and $\mathcal{I}_{base} \subseteq \mathcal{I}$ be the maximal independent sets of $(\mathcal{E}, \mathcal{I})$.

**Theorem 5.2.** *Assume the greedy pair selection phase of TCEM receives as input a sample $\mathcal{R}$ of random RRP-sets such that*

$$|nF_{\mathcal{R}}(X) - \mathbb{E}[g(X)]| < \frac{\epsilon}{2}\text{OPT} \tag{2}$$

*holds for any $X \in \mathcal{I}_{base}$ with probability at least $1 - n^{-\ell}/|\mathcal{I}_{base}|$. Then, the algorithm TCEM returns an $\left(\frac{1}{(1+2\lceil\frac{k_b}{k_r}\rceil)k_r} - \epsilon\right)$-approximate solution to the CoEM problem, with probability at least $1 - n^{-\ell}$, and runs in time $\mathcal{O}(\sum_{R \in \mathcal{R}}|R|)$.*

**Lemma 5.3.** *The size of $\mathcal{I}_{base}$ satisfies $|\mathcal{I}_{base}| \leq \binom{n}{k_r(\tau+1)}\frac{(k_r(\tau+1))!}{k_r!\,(\tau!)^{k_r}}$, where $\tau = \lceil\frac{k_b}{k_r}\rceil$. The bound is tight when $k_b \bmod k_r = 0$.*

Let $\lambda = \frac{4n}{\epsilon^2}\left(\frac{\epsilon}{3} + 2\right)(\ell \ln n + ln2 + \ln|\mathcal{I}_{base}|)$ We now give a lower bound on the size of $\mathcal{R}$ so that Equation (2) holds for all $X \in \mathcal{I}_{base}$.

**Lemma 5.4.** *Let $\mathcal{R}$ be such that $|\mathcal{R}| \geq \frac{\lambda}{\text{OPT}}$. Then, Equation (2) holds for all $X \in \mathcal{I}_{base}$ with probability at least $1 - n^{-\ell}$.*

From the previous lemma, $\mathcal{R}$ should satisfy $|\mathcal{R}| \geq \lambda/\text{OPT}$, however OPT is unknown and **NP**-hard to compute. To circumvent this problem, we follow the approach employed in previous work [3, 40] and exploit a connection to the martingale theory to adaptively estimate a lower bound of OPT by performing a statistical test $B(y)$. We perform the test iteratively on $\mathcal{O}(\log_2 n)$ values of $y = n/2, n/4, \ldots, 1$, such that, if OPT $< y$ then $B(y) = \texttt{false}$.

We now give the details of the sampling phase of TCEM which, by employing the statistical test, identifies a lower bound LB and generates the final sample on which the approximate greedy solution $\tilde{X}^G$ will be computed. The algorithm starts by initializing $\mathcal{R} = \emptyset$, a less stringent error parameter $\epsilon_2 \geq \epsilon$, and a naïve lower bound LB $= 1$. Then, it enters a $\texttt{for}$-loop with at most $\log_2 n$ iterations. In the $i$-th iteration, the algorithm computes $y = n/2^i$ and derives

$$\theta_i = \frac{1}{\epsilon_2^2}\left(\frac{2\epsilon_2}{3} + 2\right)(\ell \ln n + \ln \log_2 n + \ln|\mathcal{I}_{base}|)\frac{n}{y}.$$

Then the algorithm inserts more random RRP-sets into $\mathcal{R}$ until $|\mathcal{R}| \geq \theta_i$ and greedily computes a solution $\tilde{X}_i^G$ on this sample. If $\mathcal{R}$ satisfies the following *stopping condition*

$$nF_{\mathcal{R}}(\tilde{X}_i^G) \geq (1 + \epsilon_2)\, y, \tag{3}$$

the algorithm sets LB $= \frac{nF_{\mathcal{R}}(\tilde{X}_i^G)}{1+\epsilon_2}$ and terminates the `for`-loop. If this is the case, the algorithm generates more random RRP-sets into $\mathcal{R}$ until $|\mathcal{R}| \geq \lambda/\text{LB}$ and returns $\mathcal{R}$ as input to the greedy pair selection phase that computes $\tilde{X}^G$. Otherwise, the algorithm proceeds in the $(i + 1)$-th iteration. If after $\mathcal{O}(\log_2 n)$ iterations the algorithm cannot set LB, then the naïve lower bound is used. Our main result is the following.

**Theorem 5.5.** *With probability at least* $1 - n^{-\ell}$, TCEM *returns a sample* $\mathcal{R}$ *such that* $|\mathcal{R}| \geq \lambda/\text{OPT}$.

# 6 Experimental evaluation

We evaluate our method against different baselines on real-world networks. We measure the co-exposure size as a function of the available budget. We also evaluate the scalability of our method. The confidence and accuracy parameters are set to $\ell = 1$ and $\epsilon = 0.2$. Our experiments are performed on a server with a $2 \times 10$ core Xeon E5 2680 v2 2.80 GHz processor, with 256 GB memory.

**Datasets.** We use the following networks: Flixster [6], Last.FM [5], NetHEPT [13], and WikiVote [30]. Basic statistics of these networks are reported in the supplementary. For each network, we use three different methods to assign independent-cascade parameters. First, we use the *weighted-cascade model* [28], in which the probability of edge $(u, v)$ is set to $1/d(v)$ for both campaigns, where $d(v)$ is the in-degree of node $v$. The resulting networks are denoted by adding prefix _wc, e.g., Flixster_wc. As second and third, following [25], we devise homogeneous and heterogeneous setting using arbitrary edge probabilities: campaign-specific propagation probabilities on an edge are set to be equal in the former while they differ in the latter setting. For all the networks except Flixster, we use the *trivalency model* [14] to assign the homogeneous and heterogeneous edge probabilities by drawing randomly from $\{0.1, 0.01, 0.001\}$ for each edge. For Flixster, we use probabilities learned in [6]. The resulting networks are denoted by adding the prefixes _het and _hom, e.g., Flixster_het and Flixster_hom.

**Baselines.** We compare our method, TCEM, with four baselines: Degree-One, Degree-Two, MNI and BalanceExposure. The first two baselines consider the nodes in decreasing order of out-degree. For Degree-One, $k_r$ seeds are assigned to one campaign and $k_b$ to the other; for Degree-Two seeds are assigned in a round-robin fashion. MNI[1] solves $\arg\max_{X \in \mathcal{I}} |N'(X_r) \cap N'(X_b)|$, where $N'(X_i)$ is the union of $X_i$ and $X_i$'s out-neighbors, and $\mathcal{I}$ is given in Definition 4.1. BalanceExposure is the greedy method proposed by Garimella et al. [25], which we use without initial seeds.

**Budget selection.** Our method takes as input seed budgets $k_r$ and $k_b$, while BalanceExposure takes as input a single budget $k$, and returns optimal $k_r$ and $k_b$ for that $k$. To ensure a fair comparison, we first execute BalanceExposure with varying $k = 50, 100, 150, 200$, and use the returned values $k_r$ and $k_b$ as input for the other methods.

**Results.** Co-exposure results for different networks are shown in Figure 1. We observe that TCEM outperforms the baselines in most networks, while different independent-cascade models and network topologies have significant impact on performance. For heterogeneous and homogeneous independent-cascade models, TCEM is the best-performing method. For the weighted-cascade model, the three local algorithms that use out-degree information can perform slightly better than TCEM; this happens for instance in WikiVote_wc. This result is consistent with the empirical observation that nodes with high out-degree obtain large expected spread under the weighted-cascade model [28], resulting in large expected co-exposure values. However, such local baselines do not have robust performance as they are all outperformed by TCEM when different independent-cascade parameters are used. We observe that the BalanceExposure algorithm is consistently outperformed by all the algorithms in all the settings.

In Figure 2 we report the memory consumption and running time of TCEM, as a function of $k_r + k_b$, for the Flixster_het dataset. We fix $\tau = 2$ and $k_r = 20$. We observe that memory and time increase linearly, or better, with $k_r + k_b$. As an indicative performance result, when $k_r = 20$ and $k_b = 200$, TCEM requires 48 GB and runs in 150 minutes.

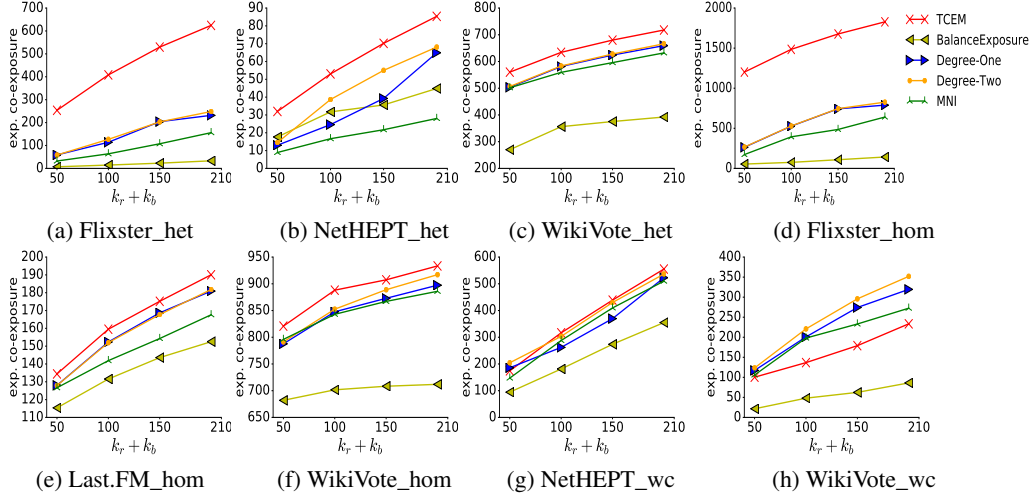

Figure 1: Co-exposure results for different networks for varying $k_r + k_b$.

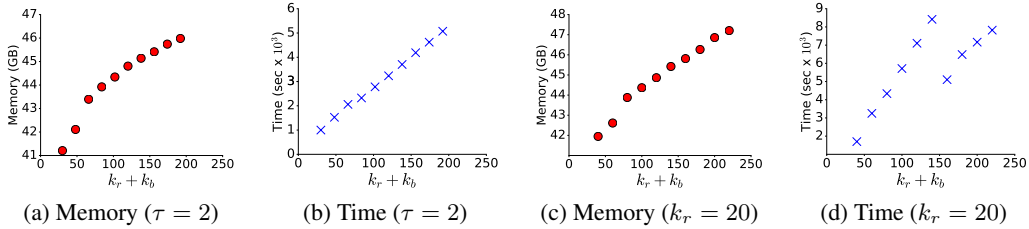

Figure 2: Memory consumption and running time of TCEM on the Flixster_het.

## 7  Conclusions

In this paper, we address the problem of maximizing co-exposure in social networks. We show that the problem is **NP**-hard and the objective function is neither submodular nor supermodular. By exploiting a connection to a submodular function that acts as a lower bound to the objective, we devise an approximation algorithm with provable guarantee. We further propose TCEM, a scalable instantiation of our approximation algorithm that can efficiently estimate the expected co-exposure.

Several directions for future work open ahead. First, it would be interesting to improve the approximation guarantee for the problem we define. Second, we would like to extend our approach to account for the social advertising setting [2] in which the advertisers, with a limited monetary budget, are required to pay a monetary amount to the host for each engagement to their virally propagating campaign.

## Acknowledgments

Part of this work was done while the authors were with Aalto University, in Finland. This research is supported by the Academy of Finland projects AIDA (317085) and MLDB (325117), the ERC Advanced Grant REBOUND (834862), the EC H2020 RIA project SoBigData (871042), and the Wallenberg AI, Autonomous Systems and Software Program (WASP) funded by the Knut and Alice Wallenberg Foundation.

## Broader impact

Our work addresses the problem of maximizing co-exposure of information in online social networks via viral-marketing strategies. We are interested in situations where opposing campaigns are propagated in different parts of a social network, with users in one side not being aware of the content and arguments seen on the other side. Although, the focus of our work is mainly theoretical, and a number of modeling considerations has been stripped out for the sake of mathematical rigor, applying this kind of ideas in practice may have significant impact towards reducing polarization on societal issues, and offering users a more balanced news diet and the possibility to participate in constructive deliberation.

On the other hand, one needs to be careful how our framework will be applied in practice. One potential source of misuse is when misinformation or disinformation is offered to counter true facts. Here we assume that this aspect is orthogonal to our approach, and that the social-network platform needs to mitigate this danger by providing mechanisms of information validation, fact checking, and ethical compliance of the content before allowing it to circulate in the network.

Another issue is that, users often do not understand why they see a particular item in their feed; the system content-filtering and prioritization algorithm is opaque to them. In the context of our proposal, since we are suggesting to make content recommendations to selected users, it is important that transparent mechanisms are in place for the users to opt in participating in such features, to understand why they receive these recommendations, and in general, to be able to control their content.

## Footnotes

[1]Short for *maximum neighborhood intersection*.

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
