[Supplementary Material]

# Co-exposure maximization in online social networks
# Supplementary material

**Sijing Tu**
Department of Computer Science
KTH Royal Institute of Technology
Stockholm, Sweden
sijing@kth.se

**Cigdem Aslay**
Department of Computer Science
Aarhus University
Aarhus, Denmark
cigdem@cs.au.dk

**Aristides Gionis**
Department of Computer Science
KTH Royal Institute of Technology
Stockholm, Sweden
argioni@kth.se

## A  Proof of Theorem 4.1

*Proof.* We prove this by using an approximation preserving reduction from the MAXIMUM COVERAGE problem. Given a universe $U = \{x_1, \ldots, x_n\}$ of $n$ elements, a collection $\mathcal{S} = \{S_1, \ldots, S_m\}$ of subsets of $U$, and an integer $k$, MAXIMUM COVERAGE problem asks to select $k$ subsets from $\mathcal{S}$ such that their union has the maximum cardinality.

Given an instance $\Pi_{\mathrm{MC}}$ of the MAXIMUM COVERAGE problem, we construct an instance $\Pi_{\mathrm{CoEM}}$ of CoEM problem as follows. First, we create a directed graph $G = (V, E)$ with the set $V$ of nodes containing the ground set $U$, a node $s_i$ for each subset $S_j \in \mathcal{S}$, and an additional node $t$, i.e., $V = \{s_1, \ldots, s_m\} \cup \{t\} \cup \{x_1, \ldots, x_n\}$. We define the set $E$ of edges as $E = \{(s_j, x_i) \mid x_i \in S_j\} \cup \{(t, x_i) \mid x_i \in U\}$. Finally, we let $k_r = 1$, $k_b = k$, and $p_{uv}^r = p_{uv}^b = 1$, for all $(u, v) \in E$.

Let $S_{\mathrm{MC}}^* = \{S_{j_1}, \ldots, S_{j_k}\}$ denote the optimal solution to MAXIMUM COVERAGE problem on the instance $\Pi_{\mathrm{MC}}$ and let $\mathrm{OPT}_{\mathrm{MC}} = |\cup_{S \in S_{\mathrm{MC}}^*} S|$. Likewise, let $(S_r^*, S_b^*)$ denote the optimal pair of seed sets maximizing the co-exposure in the instance $\Pi_{\mathrm{CoEM}}$ and let $\mathrm{OPT}_{\mathrm{CoEM}} = |I(S_r^*) \cap I(S_b^*)|$. Next, we will show that $\mathrm{OPT}_{\mathrm{MC}} = \mathrm{OPT}_{\mathrm{CoEM}}$.

First we show that $\mathrm{OPT}_{\mathrm{MC}} \leq \mathrm{OPT}_{\mathrm{CoEM}}$. Let $S_{\mathrm{MC}}^* = \{S_{j_1}, \ldots S_{j_k}\}$. Since setting $S_r = \{t\}$ and $S_b = \{s_{j_1}, \ldots s_{j_k}\}$ provides a feasible solution to CoEM, we have $\mathrm{OPT}_{\mathrm{MC}} = |U \cap (\cup_{i \in [k]} S_{j_i})| = |I(S_r) \cap I(S_b)| \leq \mathrm{OPT}_{\mathrm{CoEM}}$.

We now show that $\mathrm{OPT}_{\mathrm{CoEM}} \leq \mathrm{OPT}_{\mathrm{MC}}$. First, notice that any feasible solution $(S_r, S_b)$ to CoEM in which node $t$ is not assigned to $S_r$ is suboptimal as the number of nodes co-exposed to both campaigns would be upper bounded by the cardinality of the largest subset in $\mathcal{S}$ for such solutions. It's also easy to see that for any $S_b \in V \setminus \{t\}$ such that $S_b \cap \{x_1, \ldots, x_n\} \neq \emptyset$, we can always find another feasible $S_b'$ by replacing each $x_i \in S_b$ with a neighbor $s_j$ of $x_i$. Thus, we have $\mathrm{OPT}_{\mathrm{CoEM}} = |I(S_r^*) \cap I(S_b^*)| = |U \cap I(S_b^*)| \leq \mathrm{OPT}_{\mathrm{MC}}$.

Assume now that there is an approximation algorithm for CoEM problem with a ratio better than $1 - \frac{1}{e}$. This implies that we can also approximate the MAXIMUM COVERAGE problem with a ratio better than $1 - \frac{1}{e}$, which is a contradiction as shown by Feige et al. Feige [1998]. $\square$

## B  Proof of Lemma 4.2

*Proof.* Consider the following toy example. Let $G = (V, E)$, where $V = \{r_1, r_2, b_1, b_2, v_1, v_2\}$, and $E = \{(r_1, v_1), (r_2, v_2), (b_1, v_1), (b_2, v_2)\}$ with $p_e^i = 1$ for all $e \in E$ and $i = \{r, b\}$. Let $S_r = \{r_1\}$, $S_r' = \{r_2\}$, $S_b = \{b_1\}$, and $S_b' = \{b_2\}$. It follows that $\mathbb{E}[C(S_r, S_b)] + \mathbb{E}[C(S_r', S_b')] = 0$, while $\mathbb{E}[C(\emptyset, \emptyset)] + \mathbb{E}[C(S_r \cup S_r', S_b \cup S_b')] = 2$. which contradicts the condition of simple or directed bisubmodularity. $\square$

## C  Proof of Lemma 4.3

*Proof.* For any $(S_r, S_b) \in O_1$, we can construct a set $X \subseteq \mathcal{E}$ by pairing each node in $S_r$ with at most $\lceil \frac{k_b}{k_r} \rceil$ different nodes of $S_b$. The resulting $X$ contains $|S_b|$ pairs and is a member of $\mathcal{I}$ since it satisfies all the conditions of the set-of-pairs system $(\mathcal{E}, \mathcal{I})$. Thus, we have $O_1 \subseteq O_2$.

$\square$

## D  Proof of Lemma 4.4

*Proof.* Let $X \subseteq Y \subseteq \mathcal{E}$ and let $e \in \mathcal{E} \setminus Y$. First, we show that $f(\cdot)$ is non-decreasing. Since by definition, $X \subseteq Y$ indicates that $X_r \subseteq Y_r$ and $X_b \subseteq Y_b$, we have

$$f(X) = |I(X_r) \cap I(X_b)| \le |I(Y_r) \cap I(Y_b)| \le f(Y).$$

Next, we show that $f$ is neither submodular nor supermodular by providing counter examples on a toy graph. Let $G = (V, E)$ be a directed graph such that $V = \{r_0, r_1, b_0, b_1, b_2, v_0, v_1, v_2\}$ and $E = \{(r_0, v_0), (r_0, v_1), (r_1, v_2), (b_0, v_0), (b_0, v_1), (b_1, v_1), (b_2, v_2)\}$. Let $p_{uv}^r = p_{uv}^b = 1$, for all $(u, v) \in E$.

We first show that $f(\cdot)$ is not submodular. Let $X = \emptyset$, $Y = \{(r_0, b_2)\}$, and $e = (r_1, b_0)$. Then we have $f(X \cup \{e\}) - f(X) = 0$ while $f(Y \cup \{e\}) - f(Y) = 3$.

Now we show that $f(\cdot)$ is not supermodular. Let $X = \emptyset$, $Y = \{(r_0, b_1)\}$, $e = (r_0, b_0)$. Then we have $f(X \cup \{e\}) - f(X) = 2$, while $f(Y \cup \{e\}) - f(Y) = 0$.

$\square$

## E  Proof of Lemma 4.5

*Proof.* We prove the monotonicity and submodularity of $g(\cdot)$ over a possible world $w$ sampled from $\tilde{G} = (V, \tilde{E}, \tilde{p})$. Let $X \subseteq Y \subseteq \mathcal{E}$ and let $e = (e_r, e_b) \in \mathcal{E} \setminus Y$. We first show that $g(\cdot)$ is monotone.

$$g(X) = |\cup_{(r,b) \in X} (I(r) \cap I(b))| \le |\cup_{(r,b) \in Y} (I(r) \cap I(b))| = g(Y).$$

We now show that $g(\cdot)$ is submodular. Since $\cup_{(r,b) \in X} (I(r) \cap I(b)) \subseteq \cup_{(r,b) \in Y} (I(r) \cap I(b))$, for any $e \in \mathcal{E} \setminus Y$, it follows that

$$\begin{aligned} g(X \cup \{e\}) - g(X) &= |(I(e_r) \cap I(e_b)) \setminus \cup_{(r,b) \in X} (I(r) \cap I(b))| \\ &\ge |(I(e_r) \cap I(e_b)) \setminus \cup_{(r,b) \in Y} (I(r) \cap I(b))| \\ &= g(Y \cup \{e\}) - g(Y). \end{aligned}$$

Thus, $g$ is a non-decreasing submodular function. $\square$

## F  Proof of Lemma 4.6

*Proof.* We first prove the connection between $f$ and $g$ in any possible world $w$.

Given $X^* \subseteq \mathcal{E}$, let $(X_r^*, X_b^*)$ denote the corresponding pair of optimal seed sets. Assume wlog that $X_r^* = \{r_0, \ldots, r_{k_r - 1}\}$. Furthermore, let $\{X_{b_0}^*, \cdots, X_{b_{k_r - 1}}^*\}$ be any partitioning of $X_b^*$ into $k_r$

disjoint sets. Then we have

$$
\begin{aligned}
f(X^*) &= |I(X_r^*) \cap I(X_b^*)| = |(\cup_{i=0}^{k_r-1} I(r_i)) \cap (\cup_{j=0}^{k_r-1} I(X_{b_j}))| \\
&= |\cup_{p=0}^{k_r-1} [\cup_{i=0}^{k_r-1} (I(r_i) \cap I(X_{b_{[i+p]\%k_r}}))]| \\
&\leq k_r \, \max\{|\cup_{i=0}^{k_r-1} (I(r_i) \cap I(X_{b_{[i+0]\%k_r}}))|, \dots, |\cup_{i=0}^{k_r-1} (I(r_i) \cap I(X_{b_{[i+k_r-1]\%k_r}}))|\} \\
&\leq k_r \, g(X^0).
\end{aligned}
$$

Finally, by taking the linear combination over all possible worlds, we have $\mathbb{E}[f(X^*)] \leq k_r \, \mathbb{E}[g(X^0)]$.
$\square$

## G   Proof of Lemma 4.7

*Proof.* First, we show that $(\mathcal{E}, \mathcal{I})$ is an independence system. Let $X \in \mathcal{I}$, and let $Y$ be any set such that $Y \subseteq X$. Thus we have $Y_r \subseteq X_r$ and $Y_b \subseteq X_b$, it follows that $|Y_r| \leq |X_r| \leq k_r$; $|Y| = |Y_b| \leq |X_b| = |X| \leq k_b$ and $Y_b \cap Y_r \subseteq X_b \cap X_r = \emptyset$. Besides, for each $r_0 \in Y_r$, it follows that $\cup_{(r_0,b)\in Y}\{b\} \subseteq \cup_{(r_0,b)\in X}\{b\}$, thus $|\cup_{(r_0,b)\in Y}\{b\}| \leq \lceil \frac{k_b}{k_r} \rceil$. In conclusion, $Y \in \mathcal{I}$.

Second, $I$ is not a matroid. Let $k_r = 1$, $k_b = 2$, let $X = \{(1,2),(1,4)\}$, and $Y = \{(2,4)\}$, we have $|X| - |Y| = 1$, while neither $\{(1,2)\} \cup Y \in \mathcal{I}$ nor $\{(1,4)\} \cup Y \in \mathcal{I}$.

In conclusion, $(\mathcal{E}, \mathcal{I})$ is an independent system but not a matroid.
$\square$

## H   Proof of Lemma 4.8

*Proof.* For any $A \subseteq \mathcal{E}$, let $X$ be the maximum base of $A$, let $Y$ be the minimum base of $A$. Thus for $X$, $|X_r| \geq |X_b|/\lceil \frac{k_b}{k_r} \rceil = |X|/\lceil \frac{k_b}{k_r} \rceil$. For $Y$, $|Y_r \cup Y_b| \leq 2|Y_b| = 2|Y|$.

If $|X_r| > |Y_r| + |Y_b|$, then there is a pair that only exists in $X$, i.e. there exists $x \in X \setminus Y$, such that $\{x\} \cup Y \in I$, since both $x_r$ and $x_b$ are not in $Y_r \cup Y_b$. Thus we have $|X|/\lceil \frac{k_b}{k_r} \rceil \leq 2|Y|$, it follows that $|X|/|Y| \leq 2\lceil \frac{k_b}{k_r} \rceil$.
$\square$

## I   Proof of Theorem 4.9

*Proof.* Lemmas 4.5 and 4.8 imply that

$$
\mathbb{E}[g(X^G)] \geq \frac{1}{1 + 2\lceil \frac{k_b}{k_r} \rceil} \mathbb{E}[g(X^0)]
$$

Furthermore, by using the result in Lemma 4.6, we have

$$
\mathbb{E}[f(X_f^G)]) \geq \mathbb{E}[g(X^G)] \geq \frac{1}{1 + 2\lceil \frac{k_b}{k_r} \rceil} \mathbb{E}[g(X^0)] \geq \frac{1}{1 + 2\lceil \frac{k_b}{k_r} \rceil} \frac{\mathbb{E}[f(X^*)]}{k_r}
$$

$\square$

## J   Proof of Lemma 5.1

*Proof.* To avoid ambiguity, we use subscriptions $w$ and $v$ to denote specific samples drawn from $\tilde{G}$ and $V$, respectively; thus, if $w$ and $v$ are given, we write $g_w(X) = |\cup_{(r,b)\in X}(I_w(r) \cap I_w(b))|$, and $R_{v,w} = \{(r,b) : v \in I_w(r) \cap I_w(b)\}$.

First, it follows by definition that, in a possible world $w$, $R_{v,w} \cap X \neq \emptyset$ if and only if $\exists (r,b) \in X$ such that $v \in I_w(r) \cap I_w(b)$. Thus, in a possible world $w$, we have

$$
\begin{aligned}
g_w(X) &= |\{v \in V \mid v \in I_w(r) \cap I_w(b), (r,b) \in X\}| \\
&= |\{v \in V \mid R_{v,w} \cap X \neq \emptyset\}| \\
&= \sum_{v \in V} \mathbb{1}(R_{v,w} \cap X \neq \emptyset)
\end{aligned}
$$

where $\mathbb{1}(R_{v,w} \cap X \neq \emptyset)$ is an indicator variable that takes the value of 1 if $R_{v,w} \cap X \neq \emptyset$ and 0 otherwise. Then, we have:

$$
\begin{aligned}
\mathbb{E}[g(X)] &= \sum_{w \sqsubseteq \tilde{G}} \Pr[w] \, g_w(X) \\
&= \sum_{w \sqsubseteq \tilde{G}} \Pr[w] \sum_{v \in V} \mathbb{1}(R_{v,w} \cap X \neq \emptyset) \\
&= \sum_{v \in V} \sum_{w \sqsubseteq \tilde{G}} \Pr[w] \, \mathbb{1}(R_{v,w} \cap X \neq \emptyset) \\
&= n \, \mathbb{E}[\mathbb{1}(R \cap X \neq \emptyset)]
\end{aligned}
$$

where the last equality follows from taking the expectation over the randomness of $v \sim V$ and $w \sim \tilde{G}$.

So far we have shown that for a random RRP-set $R$, we have $\mathbb{E}[\mathbb{1}(R \cap X \neq \emptyset)] = \frac{\mathbb{E}[g(X)]}{n}$. Then, by using $F_{\mathcal{R}}(X)$ as an estimator of $\mathbb{E}[\mathbb{1}(R \cap X \neq \emptyset)]$, we have:

$$
\begin{aligned}
\mathbb{E}[F_{\mathcal{R}}(X)] &= \mathbb{E}\left[ \frac{\sum_{R \in \mathcal{R}} \mathbb{1}(R \cap X \neq \emptyset)}{|\mathcal{R}|} \right] \\
&= \frac{\sum_{R \in \mathcal{R}} \mathbb{E}[\mathbb{1}(R \cap X \neq \emptyset)]}{|\mathcal{R}|} \\
&= \frac{|\mathcal{R}| \cdot \mathbb{E}[\mathbb{1}(R \cap X \neq \emptyset)]}{|\mathcal{R}|} \\
&= \frac{\mathbb{E}[g(X)]}{n}.
\end{aligned}
$$

$\square$

## K    Proof of Theorem 5.2

We first provide the pseudocode of the greedy pair selection phase of TCEM in Algorithm 1.

---
**Algorithm 1: RR-Pairs-Greedy**

---
**input** : $\mathcal{R}, (\mathcal{E}, \mathcal{I})$
**output** : $\tilde{X}^G$

1  $\tilde{X}^G \leftarrow \emptyset$
2  $x = \arg\max_{x:\{x\} \cup \tilde{X}^G \in \mathcal{I}} F_{\mathcal{R}}(\tilde{X}^G \cup \{x\}) - F_{\mathcal{R}}(\tilde{X}^G)$
3  **while** $x \neq \emptyset$ **do**
4  $\quad$ $\tilde{X}^G = \tilde{X}^G \cup \{x\}$;
5  $\quad$ $x = \arg\max_{x:\{x\} \cup \tilde{X}^G \in \mathcal{I}} F_{\mathcal{R}}(\tilde{X}^G \cup \{x\}) - F_{\mathcal{R}}(\tilde{X}^G)$
6  **end**
7  **return** $\tilde{X}^G$

---

We now show that $F_{\mathcal{R}}(\cdot)$ is monotone. Given any $X \subset \mathcal{E}$ and $x \in \mathcal{E} \setminus X$, we have

$$
F_{\mathcal{R}}(X \cup \{x\}) = \frac{\sum_{R \in \mathcal{R}} \mathbb{1}[R \cap (X \cup \{x\}) \neq \emptyset]}{\mathcal{R}} \geq \frac{\sum_{R \in \mathcal{R}} \mathbb{1}[R \cap X \neq \emptyset]}{\mathcal{R}} = F_{\mathcal{R}}(X).
$$

Thus, $F_{\mathcal{R}}(\cdot)$ is monotone.

Next we show that $F_{\mathcal{R}}(\cdot)$ is submodular. Given any $X \subseteq Y \subset \mathcal{E}$ and $x \in \mathcal{E} \setminus Y$, we have

$$F_{\mathcal{R}}(X \cup \{x\}) - F_{\mathcal{R}}(X) = \frac{\sum_{R \in \mathcal{R}} \mathbb{1}[R \cap (X \cup \{x\}) \neq \emptyset] - \sum_{R \in \mathcal{R}} \mathbb{1}[R \cap X \neq \emptyset]}{\mathcal{R}}$$

$$= \frac{\sum_{R \in \mathcal{R}} \mathbb{1}[R \cap \{x\} \neq \emptyset, R \cap X = \emptyset]}{\mathcal{R}}$$

$$\geq \frac{\sum_{R \in \mathcal{R}} \mathbb{1}[R \cap \{x\} \neq \emptyset, R \cap Y = \emptyset]}{\mathcal{R}}$$

$$= F_{\mathcal{R}}(Y \cup \{x\}) - F_{\mathcal{R}}(Y).$$

We have shown that $F_{\mathcal{R}}(\cdot)$ is monotone and submodular. Thus, Algorithm 1 provides $\left(1 + 2\lceil \frac{k_b}{k_r} \rceil\right)$-approximation Calinescu et al. [2011] to the problem of maximizing $F_{\mathcal{R}}(X)$ on the sample $\mathcal{R}$; let $X^+ := \arg\max_{X \in \mathcal{I}} F_{\mathcal{R}}(X)$ denote the optimal solution of this problem. Then, we have

$$F_{\mathcal{R}}(\tilde{X}^G) \geq \frac{1}{1 + 2\lceil \frac{k_b}{k_r} \rceil} F_{\mathcal{R}}(X^+) \tag{1}$$

Given that $X^+$ is the optimal solution on the sample, we also have

$$F_{\mathcal{R}}(X^+) \geq F_{\mathcal{R}}(X^0) \tag{2}$$

where $X^0 = \arg\max_{X \in \mathcal{I}} \mathbb{E}[g(X)]$.

We remind that $\mathrm{OPT} = \mathbb{E}[g(X^0)]$ and that the size of the sample $\mathcal{R}$ is such that $|n F_{\mathcal{R}}(X) - \mathbb{E}[g(X)]| < \frac{\epsilon}{2}\,\mathrm{OPT}$ holds for any $X \in \mathcal{I}_{base}$ with probability at least $1 - n^{-\ell}/|\mathcal{I}_{base}|$. Then, by using Eq.s 1 and 2, and a union bound over all $n^{-\ell}/|\mathcal{I}_{base}|$ estimations, w.p. at least $1 - n^{-\ell}$ we have:

$$\mathbb{E}[g(\tilde{X}^G)] \geq n\, F_{\mathcal{R}}(\tilde{X}^G) - \frac{\epsilon}{2}\,\mathrm{OPT}$$

$$\geq \frac{1}{1 + 2\lceil \frac{k_b}{k_r} \rceil} n\, F_{\mathcal{R}}(X^+) - \frac{\epsilon}{2}\,\mathrm{OPT}$$

$$\geq \frac{1}{1 + 2\lceil \frac{k_b}{k_r} \rceil} n\, F_{\mathcal{R}}(X^0) - \frac{\epsilon}{2}\,\mathrm{OPT}$$

$$\geq \frac{1}{1 + 2\lceil \frac{k_b}{k_r} \rceil} \left(\mathbb{E}[g(X^0)] - \frac{\epsilon}{2}\,\mathrm{OPT}\right) - \frac{\epsilon}{2}\,\mathrm{OPT}$$

$$\geq \frac{1}{1 + 2\lceil \frac{k_b}{k_r} \rceil} \mathbb{E}[g(X^0)] - \epsilon\,\mathrm{OPT}$$

$$= \left(\frac{1}{1 + 2\lceil \frac{k_b}{k_r} \rceil} - \epsilon\right) \mathbb{E}[g(X^0)]$$

Finally, by using Lemma 4.6, we obtain

$$\mathbb{E}[g(\tilde{X}^G)] \geq \left(\frac{1}{1 + 2\lceil \frac{k_b}{k_r} \rceil} - \epsilon\right) \mathbb{E}[g(X^0)]$$

$$\geq \left(\frac{1}{1 + 2\lceil \frac{k_b}{k_r} \rceil} - \epsilon\right) \frac{\mathbb{E}[f(X^*)]}{k_r}$$

$$\geq \left(\frac{1}{(1 + 2\lceil \frac{k_b}{k_r} \rceil)k_r} - \epsilon\right) \mathbb{E}[f(X^*)].$$

Finally, we note that the running time of Algorithm 1 follows from the running time analysis for the maximum coverage problem; that is, it is linear in the size of the input as each pair in each RRP-set of the sample will be consider at most once, leading to $\mathcal{O}(\sum_{R \in \mathcal{R}} |R|)$.

## L  Proof of Lemma 5.3

*Proof.* For any $X \in \mathcal{I}_{base}$, we have $|X_r| = k_r$, $|X_b| = k_b$, and each node in $X_r$ is paired with $\tau$ nodes in $X_b$. Notice that, there are $\binom{n}{k_r + k_b}$ ways to select red and blue seed nodes. Once we select $k_r + k_b$ nodes, we create $k_r$ groups of at most $\tau + 1$ nodes, each of which has at most $\binom{\tau + 1}{1}$ ways to create ordered pairings by using the nodes in the group. Thus, we have

$$|\mathcal{I}_{base}| \le \binom{n}{k_r + k_b}\binom{k_r + k_b}{\tau + 1}\binom{\tau + 1}{1}\cdots\binom{k_r + k_b - (k_r - 1)(\tau + 1)}{\tau + 1}\binom{\tau + 1}{1}$$

$$= \binom{n}{k_r(\tau + 1)}\frac{(k_r(\tau + 1))!}{k_r!\,(\tau!)^{k_r}}.$$

$\square$

## M  Proof of Lemma 5.4

We follow the martingale based framework as in Tang et al. [2015], Aslay et al. [2018].

First we introduce preliminary definitions.

**Definition M.1** (Martingale). *A sequence of random variable $Y_1, Y_2, Y_3, \ldots$ is a martingale, if and only if $\mathbb{E}[|Y_i|] < +\infty$ and $\mathbb{E}[Y_i \mid Y_1, Y_2, \ldots, Y_{i-1}] = Y_{i-1}$ for any $i$.*

Given a random sample $\mathcal{R} = \{R_1, \ldots, R_\theta\}$, let $x_i$ be a binary random variable defined as $x_i = \mathbb{1}[R_i \cap X \ne \emptyset]$. By Lemma 5.1, we have $\frac{\mathbb{E}[g(X)]}{n}$. Noting that the generation of an RRP-set $R_i$ is independent of $R_1, \ldots, R_{i-1}$, we have $\mathbb{E}[x_i \mid x_1, \ldots, x_{i-1}] = \frac{\mathbb{E}[g(X)]}{n}$.

Let $x = \frac{1}{n}\mathbb{E}[g(X)]$, let $M_j = \sum_{z=1}^{j}(x_z - x)$, so $\mathbb{E}[M_j] = 0$, and

$$\mathbb{E}[M_j \mid M_1, \ldots, M_{j-1}] = \mathbb{E}[M_{j-1} + x_j - x \mid M_1, \ldots, M_{j-1}]$$
$$= M_{j-1} - x + \mathbb{E}[x_j]$$
$$= M_{j-1},$$

therefore, the sequence $M_1, \ldots, M_\theta$ is a martingale.

We have shown that $M_1, \ldots, M_\theta$ is a martingale. We now restate a concentration inequality for martingale sequences by Chung and Lu Chung and Lu [2006].

**Lemma M.1.** *[Theorem 6.1 Chung and Lu [2006]] Let $Y_1, Y_2, \ldots$ be a martingale, such that $Y_1 \le a$, $Var[Y_1] \le b_1$, $|Y_z - Y_{z-1}| \le a$ for $z \in [2, j]$, and*

$$Var[Y_z \mid Y_1, \ldots, Y_{z-1}] \le b_j, \; for \; z \in [2, j],$$

*where $Var[\cdot]$ denotes the variance. Then, for any $\gamma > 0$*

$$\Pr(Y_j - \mathbb{E}[Y_j] \ge \gamma) \le \exp\left(-\frac{\gamma^2}{2(\sum_{z=1}^{j} b_z + a\gamma/3)}\right)$$

We now use Lemma M.1 to get the concentration result for the martingale sequence $M_1, \ldots, M_\theta$. Since $x_j \in [0, 1]$ for all $j \in [1, \theta]$, we have $|M_1| = |x_1 - x| \le 1$ and $|M_j - M_{j-1}| \le 1$ for any $j \in [2, \theta]$. $Var[M_1] = Var[x_1]$, and for any $j \in [2, \theta]$

$$Var[M_j \mid M_1, \ldots, M_{j-1}] = Var[M_{j-1} + x_j - x \mid M_1, \ldots, M_{j-1}]$$
$$= Var[x_j \mid M_1, \ldots, M_{j-1}]$$
$$= Var[x_j].$$

And for $Var[x_j]$ we have that

$$Var[x_j] = \mathbb{E}[x_j^2] - \mathbb{E}[x_j]^2$$
$$= x - x^2 \le x$$

By using Lemma M.1, for $M_\theta = \sum_{j=1}^{\theta}(x_j - x)$, with $\mathbb{E}[M_\theta] = 0$, $a = 1$, $b_j = x$, for $j = 1, 2, \ldots, \theta$, and $\gamma = \delta\theta x$, we have the following corollary.

**Corollary M.1.1.** *For any $\delta > 0$,*

$$\Pr[\sum_{j=1}^{\theta} x_j - \theta x \geq \delta\theta x] \leq \exp\left(-\frac{\delta^2}{\frac{2\delta}{3}+2}\theta x\right).$$

Moreover, for the martingale $-M_1, \ldots, -M_\theta$, we similarly have $a = 1$ and $b_j = x$ for $j = 1, \ldots, \theta$. Note also that $\mathbb{E}[-M_\theta] = 0$. Hence, for $-M_\theta = \sum_{j=1}^{\theta}(x - x_j)$ and $\gamma = \delta\theta x$ we can obtain:

**Corollary M.1.2.** *For any $\delta > 0$,*

$$\Pr[\sum_{j=1}^{\theta} x_j - \theta x \leq -\delta\theta x] \leq \exp\left(-\frac{\delta^2}{\frac{2\delta}{3}+2}\theta x\right).$$

We are now ready to prove Lemma 5.4.

*Proof.* Using Corollaries M.1.1 and M.1.2 and letting $\delta = \dfrac{\epsilon\,\mathrm{OPT}}{2nx}$, we obtain

$$
\begin{aligned}
\Pr[|nF_{\mathcal{R}}(X) - \mathbb{E}[g(X)]| \geq \tfrac{\epsilon}{2}\mathrm{OPT}] &= \mathbb{P}[|\sum_{i=1}^{\theta} x_i - \theta x| \geq \frac{\theta\epsilon}{2n}OPT] \\
&\leq 2\exp\left(-\frac{\delta^2}{\frac{2\delta}{3}+2}\theta x\right) \\
&= 2\exp\left(-\frac{3\epsilon^2\,\mathrm{OPT}^2}{4\,n(\epsilon\,\mathrm{OPT}+6nx)}\theta\right) \\
&\leq 2\exp\left(-\frac{3\epsilon^2\,\mathrm{OPT}^2}{4\,n(\epsilon\,\mathrm{OPT}+6\mathrm{OPT})}\theta\right) \\
&= 2\exp\left(-\frac{\epsilon^2\,\mathrm{OPT}}{4\,n(\frac{\epsilon}{3}+2)}\theta\right),
\end{aligned}
$$

where the last inequality above follows from the fact that $nx \leq \mathrm{OPT}$. Finally, by requiring

$$2\exp\left(-\frac{\epsilon^2\,\mathrm{OPT}}{4\,n(\frac{\epsilon}{3}+2)}\theta\right) \leq \frac{1}{n^\ell\,|\mathcal{I}_{base}|},$$

we obtain the lower bound on $\theta$.

□

# N  Proof of Theorem 5.5

We provide the pseudocode of the sampling phase of TCEM in Algorithm 2.

Let $\beta = \frac{(\frac{2}{3}\epsilon_2 + 2)(l\ln n + \ln\log_2 2n + \ln|\mathcal{I}_{base}|)n}{\epsilon_2^2}$. To prove Theorem 5.5, we first prove Lemma N.1, Lemma N.2. In these lemmas, we show that we can return a lower bound of OPT with high probability.

**Lemma N.1.** *Let $\tilde{X}$ be the output of Algorithm 1, when the size of sampled $\mathcal{R}$ is $\theta$ and*

$$\theta > \frac{(\frac{2}{3}\epsilon_2 + 2)(l\ln n + \ln\log_2 2n + \ln|\mathcal{I}_{base}|)}{\epsilon_2^2}\frac{n}{y},$$

*if OPT $< y$, then $n\,F_{\mathcal{R}}(\tilde{X}) < (1 + \epsilon_2)y$, with probability at least $1 - \frac{n^{-\ell}}{\log_2 n}$.*

*Proof.* To prove this, we will show that, when OPT $< y$, the probability that $nF_{\mathcal{R}}(X) \geq (1 + \epsilon_2)\,y$ is at most $\frac{n^{-\ell}}{\log_2 n\,|\mathcal{I}_{base}|}$. Let $X$ be arbitrary $X \in \mathcal{I}_{base}$ and let $x = \frac{1}{n}\mathbb{E}[g(X)]$. Assume that OPT $< y$

---

Algorithm 2: Sampling

**Input** : $\tilde{G}, \lambda, \beta, \epsilon_2, \tilde{I}$
**Output** : $\mathcal{R}$
1     $\mathcal{R} \leftarrow \emptyset$ ;
2     LB $\leftarrow$ LB$_0$ ;
3     **for** $i = 1, \dots, \log_2 n - 1$ **do**
4        $y \leftarrow n/2^i$ ;
5        $\theta_i = \frac{\beta}{y}$ ;
6        **while** $|\mathcal{R}| \leq \theta_i$ **do**
7           $\mathcal{R} \leftarrow \mathcal{R} \cup$ GenerateRRP-Set;
8        **end**
9        $\tilde{X}_i \leftarrow$ RR-Pairs-Greedy$(\mathcal{R}, \tilde{I})$ ;
10       **if** $n\, F_{\mathcal{R}}(\tilde{X}_i) \geq (1 + \epsilon_2)\, y$, **then**
11           LB $\leftarrow \frac{n\, F_{\mathcal{R}}(\tilde{X}_i)}{1 + \epsilon_2}$ ;
12           **break**;
13        **end**
14     **end**
15     $\theta \leftarrow \lambda/$LB;
16     **while** $|\mathcal{R}| \leq \theta$ **do**
17        $\mathcal{R} \leftarrow \mathcal{R} \cup$ GenerateRRP-Set ;
18     **end**
19     **Return** $\mathcal{R}$;

---

which implies that $x < \frac{\text{OPT}}{n} < \frac{y}{n}$, and $1 < \frac{y}{xn}$. Notice that by construction $y \leq n$ since $y \leftarrow n/2^i$. Then, by using Corollary M.1.1, we have

$$
\begin{aligned}
\Pr[nF_{\mathcal{R}}(X) \geq (1+\epsilon)y] &= \Pr\left[\theta F_{\mathcal{R}}(X) - \theta x \geq \theta x \left(\frac{(1+\epsilon_2)y}{nx} - 1\right)\right] \\
&\leq \Pr[\theta F_{\mathcal{R}}(X) - \theta x \geq \theta x \epsilon_2] \\
&\leq \exp\left(-\frac{\epsilon_2^2}{\frac{2}{3}\epsilon_2 + 2}\theta\right) \\
&\leq \exp\left(-\frac{\epsilon_2^2}{\frac{2}{3}\epsilon_2 + 2}\frac{y}{n}\theta\right) \\
&\leq \frac{n^{-\ell}}{\log_2 n |\mathcal{I}_{base}|}
\end{aligned}
$$

Finally by a union bound, we conclude that if OPT $< y$, then $nF_{\mathcal{R}}(\tilde{X}) < (1 + \epsilon_2)y$ w.p. at least $1 - \frac{n^{-\ell}}{\log_2 n}$. $\qquad\square$

**Lemma N.2.** *Let $\tilde{X}$ be the output of Algorithm 1, when the size of sampled $\mathcal{R}$ is $\theta$ and*

$$
\theta > \frac{(\frac{2}{3}\epsilon_2 + 2)(l \ln n + \ln \log_2 2n + \ln|\mathcal{I}_{base}|)}{\epsilon_2^2}\frac{n}{y},
$$

*if* OPT $\geq y$, *then* $n\, F_{\mathcal{R}}(\tilde{X}) \leq (1 + \epsilon_2)$OPT *with probability at least* $1 - \frac{n^{-\ell}}{\log_2(n)}$.

*Proof.* Let $X$ be arbitrary $X \in \mathcal{I}_{base}$. Assume that OPT $\geq y$, let $x = \frac{1}{n}\mathbb{E}[g(X)]$, thus $\frac{\text{OPT}}{nx} \geq 1$. We will now show that when OPT $\geq y$, the probability that $n\, F_{\mathcal{R}}(\tilde{X}) > (1 + \epsilon_2)$OPT is at most

$\frac{n^{-\ell}}{\log_2 n |\mathcal{I}_{base}|}$. By using Corollary M.1.2, we obtain

$$\mathbb{P}[n\,F_{\mathcal{R}}(X) > (1+\epsilon_2)\text{OPT}] = \Pr\left[\theta F_{\mathcal{R}}(X) - \theta x > \theta x\left(\frac{(1+\epsilon_2)\text{OPT}}{nx} - 1\right)\right]$$
$$\leq \exp\left(-\frac{\epsilon_2^2}{\frac{2}{3}\epsilon_2 + 2}\frac{y}{n}\theta\right)$$
$$\leq \frac{n^{-\ell}}{\log_2 n |\mathcal{I}_{base}|}$$

By taking a union bound, we reach the desired result. $\qquad\square$

Now we prove Theorem 5.5.

*Proof.* Let $i^* = \lceil \log_2 \frac{n}{\text{OPT}} \rceil$. We will first show that the probability the stopping condition holds while OPT $< y$ is at most $(i^* - 1)/(n^\ell \log_2 n)$. Recall that the value of $y$ is determined by $n/2^i$ at each iteration $i$. Then for any $i < i^*$, we have $y = n/2^i <$ OPT. Thus, by Lemma N.1 and the union bound over $i^* - 1$ iterations, the probability that OPT $< y$ and $nF_{\mathcal{R}}(X) \geq (1+\epsilon_2)y$ is at most $\frac{i^*-1}{n^\ell \log_2 n}$. Furthermore, it follows from Lemma N.2 that the probability that OPT $\geq y$ and $n\,F_{\mathcal{R}}(\tilde{X}) > (1+\epsilon_2)\text{OPT}$ is at most $1/(n^\ell \log_2 n)$. Hence, when the stopping condition holds, by union bound, the probability that OPT $\geq y$ and $n\,F_{\mathcal{R}}(\tilde{X}) \leq (1+\epsilon_2)\text{OPT}$ is at least

$$1 - \left(\frac{i^*-1}{n^\ell \log_2 n} + \frac{1}{n^\ell \log_2 n}\right) \geq 1 - n^{-\ell}.$$

Then, by Lemma N.2 and the union bound, it follows that w.p. at least $1 - n^{-\ell}$, we have

$$\text{OPT} \geq \frac{n\,F_{\mathcal{R}}(\tilde{X})}{1+\epsilon} \geq y.$$

Therefore, the algorithm sets LB$\leq$OPT w.p. at least $1 - n^{-\ell}$ and returns a sample $\mathcal{R}$ such that

$$|\mathcal{R}| \geq \frac{\lambda}{\text{LB}} \geq \frac{\lambda}{\text{OPT}}$$

w.p. at least $1 - n^{-\ell}$. $\qquad\square$

# O Experiment

## O.1 Experiment Setup

In Table 1, we report the graph statistics on number of nodes, edges and average degree.

Table 1: Statistics of the Datasets

| Dataset | $n$ | $m$ | $d(G)$ |
|---------|------|--------|--------|
| Flixster | 28843 | 272786 | 9.4576 |
| Last.FM | 1372 | 14708 | 10.72 |
| NetHEPT | 15229 | 62752 | 4.1204 |
| WikiVote | 7115 | 103689 | 14.57 |

**Baseline**. MNI[1] solves $\arg\max_{X \in \mathcal{I}} |N'(X_r) \cap N'(X_b)|$, where $N'(X_i)$ is the union of $X_i$ and $X_i$'s out neighbors, and $\mathcal{I}$ is given in Definition 3.1. MNI has a similar formulation as the $f$; while MNI only picks the seed nodes that can maximally influence the neighbors without taking further nodes into consideration. We apply this local-based method as a comparison with our global based algorithm TCEM.

## O.2 results

We set $k_r = 20$, and $k_b = 20 : 10 : 200$, here we use the Matlab grammar for list, e.g. $k_r = 20 : 10 : 200$ means $k_r$ ranges from 20 to 200 with increment value 10. The results are put in Figure 1.

Figure 1: Co-exposure results for different networks for varying $k_b$.

We set $k_b = k_r$, and $k_r = 10 : 6 : 64$. The results are put in Figure 2.

(a) Flixster_het     (b) Last.FM_het     (c) NetHEPT_het     (d) WikiVote_het

(e) Flixster_hom     (f) Last.FM_hom     (g) NetHEPT_hom     (h) WikiVote_hom

(i) Flixster_wc     (j) Last.FM_wc     (k) NetHEPT_wc     (l) WikiVote_wc

Figure 2: Co-exposure results comparison on fixed $k_b = k_r$

We set $k_r = 10 : 6 : 64$, and $k_b = 1.5k_r$. The results are put in Figure 3.

(a) Flixster_het    (b) Last.FM_het    (c) NetHEPT_het    (d) WikiVote_het

(e) Flixster_hom    (f) Last.FM_hom    (g) NetHEPT_hom    (h) WikiVote_hom

(i) Flixster_wc    (j) Last.FM_wc    (k) NetHEPT_wc    (l) WikiVote_wc

Figure 3: Co-exposure results comparison on fixed $k_b = 1.5k_r$

We set $k_r = 10 : 6 : 64$, and $k_b = 2k_r$. The results are put in Figure 5.

Figure 4: Performance comparison with fixed $\tau = 2$

We set $k_r + k_b = 50, 100, 150, 200$, we first obtain the $k_r$ and $k_b$ through implementing BalanceExposure, then we compare the performances of all the algorithms.

(a) Flixster_het     (b) Last.FM_het     (c) NetHEPT_het     (d) WikiVote_het

(e) Flixster_hom     (f) Last.FM_hom     (g) NetHEPT_hom     (h) WikiVote_hom

(i) Flixster_wc     (j) Last.FM_wc     (k) NetHEPT_wc     (l) WikiVote_wc

Figure 5: Performance comparison with fixed $k_r + k_b$

## O.3 Time and Memory

We set $k_r = 20$, and $k_b = 20 : 10 : 200$.We show the Memory and Time consumption with $k_r + k_b$ increasing. The results are put in Figure 6, and Figure 7.

(a) Flixster_het     (b) Last.FM_het     (c) NetHEPT_het     (d) WikiVote_het

(e) Flixster_hom     (f) Last.FM_hom     (g) NetHEPT_hom     (h) WikiVote_hom

(i) Flixster_wc     (j) Last.FM_wc     (k) NetHEPT_wc     (l) WikiVote_wc

Figure 6: Memory consumption with varying $k_r + k_b$.

(a) Flixster_het     (b) Last.FM_het     (c) NetHEPT_het     (d) WikiVote_het

(e) Flixster_hom     (f) Last.FM_hom     (g) NetHEPT_hom     (h) WikiVote_hom

(i) Flixster_wc     (j) Last.FM_wc     (k) NetHEPT_wc     (l) WikiVote_wc

Figure 7: Time consumption with varying $k_r + k_b$.

## Footnotes

[1]Short for *maximum neighborhood intersection*.