[Reviews · NeurIPS 2020]

Review 1

Summary and Contributions: The paper is motivated by the opinion-polarization issue (e.g., in political discussion). It presents a novel objective for content recommendation algorithms with a goal of exposing users to arguments of all competing parties, thereby potentially reducing polarization. The paper provides a rigorous analysis of the problem and designs an algorithm that can approximate the optimal solution of the optimization problem. Finally, the algorithm is evaluated in a series of semi-synthetic experiments.

Strengths: 1. Motivation and significance of the broad problem studied in the paper. I find the problem studied in the paper well-motivated, important and interesting to the broad ML community. Indeed, the algorithmically-curated content can impact opinions of many members of society in a strong way and principled design of such algorithms is required. 2.Theoretical investigation. The paper provides an in-depth treatment of the problem. It first establishes a hardness result and then uses a careful construction to design an algorithm that provably derives an approximation of the optimal solution. I should caveat, however, that I have not carefully studied the proofs of the results. Please also see related comments in the weakness part. 3. Empirical evaluation. The paper uses semi-synthetic experiments (real network structure, but simulated propagation) to compare the proposed algorithms with several baselines and shows that the new algorithm compares favourably to the baselines. Please see the weakness section for some qualification to the above point.

Weaknesses: 1. Motivation of the co-exposure maximisation formulation. It seems to me that each campaigner would rather be interested in getting as much users exposed to their campaign as possible and it is the host who may be interested in allocating additional seeds to ensure co-exposure. In fact, it appears that [25] does exactly this: they consider two sets as given and look for an additional set of nodes to balance. I think the paper would benefit from better motivating the proposed formulation (Problem 2.1.). Does it come from a merely theoretical interest or there is some realistic use case when this formulation is better than that of [25]. Additionally, Problem 2.1 seems to assume that campaigns come to the host simultaneously. Is this a realistic assumption (seems pretty strong to me)? Again, from that point of view, [25] appears to be somewhat more practical as it allows for post-hoc balancing (i.e., campaigns do not have to come in parallel). 2. Novelty. First, looking at the past literature [3, 25] it seems to me that motivation for the problem is not entirely novel and the paper continues the line of the past works on the issue. Specifically, the intro closely follows along the intro of [3]. Second, while I have not closely inspected theoretical analysis, I note that there seem to be some intersection with [3]. In particular, in lines 222--225 authors say that they "introduce a non-trivial generalisation of RR-sets called RRP". However, authors of [3] also say that they introduce a non-trivial generalization of RR-sets called RC. The paper could perhaps comment on how the approach towards designing RRP is different from the approach towards designing RC. Coupled with the above concern about the motivation of the specific theoretical formulation considered in the paper, I'm concerned about novelty and significance of the work.

Correctness: I have not carefully verified the proofs, but from the brief look the methodology appears to be sound. The experiments performed in the paper are reasonable and are designed to be as realistic as possible.

Clarity: Yes, the paper is well-written. One suggestion: maybe the writing of sections 3 and 4 can be improved by placing some accents on what are the key result and what are auxiliary. Maybe lemmas can be delegated to the appendix and instead more high-level discussion is added?

Relation to Prior Work: I think that the distinction from past work is not clear. Conceptually, the paper follows the line of past works with the difference of introducing a different objective, that is not well-motivated. On a theory side, I would appreciate a more clear discussion of difference from the past work (e.g., [3]).

Reproducibility: Yes

Additional Feedback: I would like to ensure authors that I'm looking towards their response and open to reconsidering my evaluation. To simplify the response, below are the concrete questions that I'd like to be addressed: 1. What is the practical motivation for Problem 2.1 vs. that of [25]? Isn't the assumption of campaigns being initiated in parallel a strong assumption? 2. Does construction of the RRP set conceptually differ from construction of the RC set in [25]? -------POST REBUTTAL-------- I thank authors for detailed response to my questions. In light of the response and subsequent discussion, I am happy to increase my score. I think the paper presents an important and novel work that is of interest to NeurIPS community.


Review 2

Summary and Contributions: Contributions of this paper are summarized below: * Propose Co-Exposure Maximization (CoEM), a combinatorial optimization problem asking to select two fixed-size disjoint seed sets so that the expected number of nodes who are co-exposed to both two campaigns during the diffusion process is maximized. * Prove that the objective function is neither submodular nor supermodular, where we cannot the standard greedy algorithm. * Develop an approximation algorithm (Pairs-Greedy) for CoEM, that approximates the optimal solution within a factor of (roughly) 1/(kr+2*kb), based on a technique for bounding the objective from below by a submodular function and submodular optimization on p-systems. * Introduce a speeding-up technique (TCEM) by extending the notion of Reverse Reachable sets, while guaranteeing solution quality. * Conduct experimental evaluations to demonstrate that TCEM performs better than baselines, such as BalanceExposure and naive heuristics.

Strengths: * The problem formulation is very-simple, but well-motivated from the recent emerging concerns about societal polarization, echo chambers, and filter bubbles. * The CoEM is a natural extension of original Influence Maximization, wherein we measure the number of nodes activated in both campaigns simultaneously, and it is an interesting observation that the objective function is neither submodular nor supermodular, while many extensions in relevant works result in a submodular objective, which can be easily optimized through the greedy algorithm. * Bounding the objective function by a (different) submodular function and making the use of p-systems to express the constraint is a good, certainly new idea. Overall, I like the simplicity of the definition of CoEM while capturing the essence of co-exposure, and the algorithmic development exploiting nontrivial ideas and techniques.

Weaknesses: * Approximation guarantee of the proposed algorithm (Pairs-Greedy/TCEM) for CoEM looks a bit weak: Theorem 3.8 guarantees that Pairs-Greedy returns an approximate solution within a factor of 1/O(kr+kb). This factor obviously depends on kr and kb, which is restrictive compared to an immediate inapproximability result of a (1-1/e)-factor (Theorem 3.1). Also, since kr+kb>=50 in experiments, this factor seems not to give a meaningful guarantee in practice. Further theoretical analyses to support the approximation guarantee of Pairs-Greedy (e.g., a tight inapproximability result) can be expected. * Classical greedy could be testable in experimental evaluation. While the authors compare TCEM to four baseline algorithms (i.e., BalanceExposure, Degree-One, Degree-Two, MNI), BalanceExposure is designed to optimize a different objective from CoEM, and the last three methods even do not consider the CoEM objective or the network diffusion process. I feel that a natural competitor to Pairs-Greedy/TCEM is a standard greedy algorithm, which, for example, iteratively selects a single node (with either red color or blue color) that achieves the maximum marginal increase in the objective function. Though the greedy seems not to have any approximation guarantee as discussed in the paper (e.g., submodularity ratio = 0), the greedy algorithm often works better in practice. I would like to see such a comparison of Pairs-Greedy/TCEM to the greedy algorithm, which would strengthen the superiority of the proposed solution over an obvious choice of greedy.

Correctness: The proofs of claims in the paper sound correct. In particular, I have verified the correctness of the proofs of Lemmas and Theorems in Section 3, which analyze the mathematical properties of CoEM and are essential in devising the algorithm (Pairs-Greedy/TCEM).

Clarity: This paper is generally well-written and easy-to-follow. Typos: * Proof of Lemma 3.5 in Appendix E: Notation "||" for set size is missing, and some "k" should be "k_r".

Relation to Prior Work: The definition of CoEM is reminiscent of bisubomdular functions or k-submodular functions in general; please refer to the articles below. [Singh-Guillory-Bilmes. AISTATS'12. On bisubmodular maximization] [Ohsaka-Yoshida. NIPS'15. Monotone k-Submodular Function Maximization with Size Constraints] In particular, Ohsaka & Yoshida consider an extension of Influence Maximization, wherein we can distribute one among k kinds of items to each user (with limited budget), and the objective is to maximize the number of nodes who are exposed to *at least one* of the campaigns, and prove that the greedy algorithm is 1/3-approximation. This definition is very similar to CoEM; does CoEM fit into bisubmodular/k-submodular frameworks, or how can CoEM be differentiated from such an extension?

Reproducibility: Yes

Additional Feedback: --------AUTHOR FEEDBACK-------- I appreciate the authors' feedback. I acknowledge that further improving (in)approximability is somewhat hard and the current result is acceptable. I keep the same score.


Review 3

Summary and Contributions: In this paper, the authors dealt with the problem of influence maximization through social network with a constraint. In particular, the authors' objective is to allocate seed users to two campaigns with the aim to maximize the expected number of users who are co-exposed to both campaigns. The authors then conducted theoretical analysis of this problem and proposed efficient approximation algorithms as it is NP-hard.

Strengths: 1. The problem of co-exposure maximization is interesting. 2. The theoretical analysis is sound.

Weaknesses: 1. The proposed propagation model is weak 2. Several less realistic assumptions made in the paper

Correctness: Not really.

Clarity: Not really. Its just moderately written paper

Relation to Prior Work: Not all relevant related prior literature is cited.

Reproducibility: Yes

Additional Feedback: Below are a few comments on this paper: (1) Some important and relevant references are missing in this paper such as: such as https://cs.ucsb.edu9/sites/default/files/docs/reports/2010-02.pdf The coverage of relevant literature and positioning of the work should have been better. (2) There is no real world scenario motivating scenario wherein both the competing campaigns approach the same central authority to assign seeds? (3) Why the seed sets to be disjoint? An individual can have experience of both positive and negative opinions in reality. And further he can be carrier for both such opinions. (4) In Page 2, why the propagation of both the campaigns should follow the same model? It appears to be very unrealistic and sounds like a hypothetical scenario. (5) Regarding the proposed Propagation model in Page 2 & 3, is there any data based evidence for this sort of model? Why only this model why not some other one? What is the motivation for having the proposed propagation model? This model is not convincing.


Review 4

Summary and Contributions: The paper considers an extension to the influence maximization problem where two campaigns co-exist in the network and the objective is to allocate a set of seed-nodes to each campaign, under cardinality constraints, such that the expected number of nodes that are exposed to both campaigns is maximized. The diffusion model is assumed to be the Independent Cascade model.  The problem is well-motivated, as it is clear that the problem of polarization exists in social networks and media, and the network owners have incentives to take action in reducing the polarization among their users. The theoretical findings in the paper are very interesting and novel, and the ideas explored can be applicable to other problems as well. I enjoyed reading the paper. The authors first formally define the optimization problem, and show that it is NP-hard and NP-hard to approximate within an (1-1/e) factor. Then, they show that the usual roadmap (showing submodularity and monotonicity and applying greedy) does not apply to their objective, as their optimization function is neither submodular non monotone and in fact has submodularity ratio 0. To overcome this barrier, the authors formulate a related problem and show that optimizing the related problems (which turns out to be submodular and monotone) leads to a bounded approximation for the original problem considered in this paper. The formulated problem can essentially be thought as  the function where one views the insertion of two nodes as an insertion of one node, in terms of the nodes they influence, i.e., a pair of nodes r,b influences only the nodes in the intersection of the influence set of each of r,b. The newly formulated problem is not as easy to solve as one does not simply choose arbitrary pairs of nodes and has to respect the cardinalities constraints of the original problem, which causes a greedy algorithm to introduce another multiplicative factor (that depends on the ratio of the cardinalities for the two campaigns) in the approximation that they get for the original problem. Further, the authors consider efficient implementations of their greedy algorithm. To that end, they extend the notion of sampling reverse reachable sets that has been successfully used for speeding up the greedy approaches in the traditional variants of the influence maximization problem. These sets basically provide an efficient way to estimate the number of nodes that each pair of vertices can reach. The authors provide an upper bound on the number of samples that ones need to perform in order to speed up their greedy algorithm and at the same time guarantee a good approximation, which holds with high probability. While the extension is easy to grasp, the analysis requires a lot of work.  The experimental part contains thorough experiments comparing the proposed method with several baselines and also to the work that is the most similar to the problem studied in this work, i.e., reference [25]. The objective in [25] is slightly different, in the sense that it also gets credit for nodes that are not exposed to any campaign (one optimum solution is to assign no seed nodes). Nonetheless, this seems to be a reasonable comparison. The experiments suggest that the newly proposed algorithm outperforms all other competitors in most cases, and in fact performs much better than the algorithm in [25]. While the algorithm is outperformed in a few instances, the heuristics that perform better do not provide theoretical guarantees and are known to exhibit unstable behavior. Finally, the authors also demonstrate that their algorithm scales well when increasing k_r+k_b. One arguable weakness of the algorithm is its scalability. While it scales well when increasing k_r+k_b, it seems that the computation would take several days when applied to very large graphs.

Strengths: Strong and interesting theoretical findings Novel contributions Well-motivated problem

Weaknesses: Interesting, but not a traditional machine-learning paper. The scalability seems limited. The proposed method is probably not realistically applicable to networks that have hundreds of millions of nodes and edges.

Correctness: I do not see any flaw in the theoretical claims (didn't go through the whole appending), and the experiments reflect a reasonable effort to assess the performance of the proposed method.

Clarity: Yes, the writing is another strong point of this paper.

Relation to Prior Work: Yes, prior work is discussed, and the most similar papers are further explained.

Reproducibility: Yes

Additional Feedback: Could you elaborate a bit more on the randomized BFS that is used for generating RRPs, or provide a reference? While I can guess what you mean, I think it would help to be more specific. In the running time you state in Theorem 4.2, don't you need a multiplicative factor (k_b+k_r) to account for the number of rounds of greedy? It seems possible to derive the running time of the algorithm in terms of n and OPT. While I understand that this is not representative as the RRPs are generated via a stochastic process, it would be useful to get a sense on how much does the running time improves compared to MC simulations. I would be OK if you plug in also a factor r, which is the expected size of the RRPs. In the experiments, how do you explain that in some experiments adding (k_b+k_r) nodes leads to an increase of less than (k_b + k_r) in the expected number of influenced nodes, e.g., for NetHERT_het? Do the main theorems work for the case where there are more than two campaigns, say three or four? I guess the running time of the algorithm gets prohibitively large, by I believe it would be an interesting extension. Nonetheless, consider stating this as an open problem.

[Author Response · NeurIPS 2020]

## Reply to Reviewer #1

– *Practical motivation of our problem and comparison with [25]:* We would first like to remind that campaigns being initiated simultaneously by a social-network owner is a standard assumption in the literature on influence maximization with multiple campaigns [1,12,20,29]. We believe that this assumption is even more valid in the current era, with social-media companies being under public and legal scrutiny, requiring them to make more informed and transparent decisions. Still, we acknowledge that such assumption could reduce the generality of the problem; allowing initial seeds, selected at the prior rounds of the campaigning process, would constitute an interesting future work. Such an extension, however, would require to study the problem in an online and adaptive manner (along the axis of exploration vs. exploitation) while [25] simplifies the problem to a great extent and assumes that four different seed sets are all initiated in parallel. We should note that [25] maximizes a different objective, which we believe is somewhat artificial to reduce polarization, i.e., it also accounts for nodes *not being informed* by any of the campaigns, while co-exposure maximization is a natural objective, solely accounting for nodes that are informed by both sides of a controversial issue. Besides, among the three algorithms that [25] proposes, only one provides an approximation guarantee. Finally, their algorithms rely on choosing common seeds, hence, are not applicable when the seeds sets are required to be disjoint.

– *Conceptual differences between RRP set and RC set in [3]:* Both [3] and our algorithm TCEM extend the framework of IMM [38]. Therefore, there are high-level similarities. However, since [3] and TCEM solve different problems, the way these algorithms exploit reverse-reachability and their sample-size requirements differ greatly from each other. Notice that the domain where the random sets are sampled from is dictated by the objective function: [3] operates by sampling RC sets, i.e., random sets defined over user-item pairs, while TCEM samples RRP sets, i.e., random sets defined over user-user pairs. As these algorithms sample from different domains, the sample-complexity results and the sample of random sets obtained for one problem cannot be used to solve the other.

To further elaborate the differences of our work from [3], their problem assumes as input political leanings of nodes and items, quantified in the interval $[-1, 1]$, and aims to maximize the sum, over all nodes, of the range of the exposed political leanings including the node's own. Therefore, users who are exposed to only one item that has a different leaning from theirs, still contribute to the value of their objective. Translating this to our setting by assuming two items of leaning -1 and 1, implies that the objective of [3] can potentially achieve a relatively high value while the co-exposure being 0. Thus, the work in [3] does not guarantee that co-exposure is maximized. We thank the reviewer for their constructive feedback; we will clarify these points in the paper.

## Reply to Reviewer #2

– *Weak approximation guarantee:* We acknowledge that obtaining a tighter guarantee is a highly interesting future work. We remind that our problem is not (bi-)submodular and has a submodularity ratio of 0, rendering the recent advances in monotone non-submodular function maximization non-applicable. Thus, we believe that our current result is theoretically interesting.

– *Round-robin greedy as a baseline:* We thank the reviewer for this suggestion and will consider it.

– *Bisubmodular or $k$-submodular functions:* our objective function is not bisubmodular. A simple counter-example is omitted due to space limitations. We will add this discussion in the paper.

## Reply to Reviewer #3

– *Missing references and a pointer to such a reference:* The mentioned paper studies the problem of limiting misinformation, thus, is not related to the problem studied in our paper.

– *Real-world motivating scenarios for our work:* We would like to point out that competing campaigns being initiated by a social-network owner is a standard scenario in the viral-marketing literature [1,12,20,29] since it follows the real-world business model of social-network owners, such as, Facebook and Twitter, which provide social advertising service to marketers.

– *Disjoint seed sets:* Disjointness constraint is a widely adopted and credible design choice as an opinion leader would not be promoting two sides of a controversial issue, such as, gun control. We refer the reviewer to a WSDM2018 tutorial "Influence maximization in online social networks," and references therein.

– *Propagation model:* The independent-cascade (IC) model [28], is a widely-adopted information-propagation model in the literature, e.g., see the WSDM2018 tutorial. Please note also that campaigns operate on different IC model instances.

## Reply to Reviewer #4

– *Scalability:* We remind that we are dealing with a more stringent estimation task than of the influence-maximization problem. This naturally translates to increased sample complexity, hence, less scalability compared to IMM [25]. In practise, our algorithm could scale up by a parallel implementation since RRP sets can be sampled independently.

[Meta-Review · NeurIPS 2020]

Three reviewers (two of which have high confidence) are in favor of accepting the paper. I agree with them that the paper is interesting, and would be a good addition to the program.